# Evaluating the health impact, health-system costs and cost-effectiveness of using TrueNat on stool samples compared to usual care for the diagnosis of paediatric tuberculosis in primary care settings: A modelling analysis

Nyashadzaishe Mafirakureva[1*‡], Olugbenga Kayode Daniel[2‡], Olabamiji Jamiu Olayinka[2], Kingsley Chinedum Ochei[3], Eveline Klinkenberg[4], Austin Ihesie[3], Debby Nongo[3], Rupert Amanze Eneogu[3], Andwele Mwansasu[5], Emeka Uga Elom[6], Agbaje Vivian Aderonke[2], Patrick Sunday Dakum[2], Charles Olalekan Mensah[2], Oluwafemi Christopher Odola[2], Abiola Oladotun Olayemi[2], Emily Yemisi Faleye[7], Adekunle Omotoso Makinde[8], Peter J. Dodd[1]

**1** Sheffield Centre for Health and Related Research, School of Medicine and Population Health, University of Sheffield, Sheffield, United Kingdom, **2** Institute of Human Virology, Abuja, Nigeria, **3** USAID, Abuja, Nigeria, **4** Independent Consultant, Connect TB, The Hague, The Netherlands, **5** Infectious Disease Detection and Surveillance, Rockville, Maryland, United States of America, **6** Federal Ministry of Health, National Tuberculosis, Buruli Ulcer & Leprosy Control Programme, Abuja, Nigeria, **7** Osun State Ministry of Health, State Tuberculosis, Buruli Ulcer & Leprosy Control Programme, Osogbo, Nigeria, **8** Oyo State Ministry of Health, State Tuberculosis, Buruli Ulcer & Leprosy Control Programme, Ibadan, Nigeria

‡ These authors share co-authors on this work.
* n.mafirakureva@sheffield.ac.uk

## Abstract

The World Health Organisation (WHO) recommends rapid molecular diagnostics to improve bacteriological confirmation of tuberculosis in children. TrueNat MTB, MTB Plus and MTB-RIF Dx assays (Molbio Diagnostics, India), recommended by WHO, hold potential as point-of-care tests in resource-limited settings. Using stool samples with these assays could enhance testing access, improve linkage to care, reduce costs, and increase cost-effectiveness over traditional methods. However, evidence on their costs and cost-effectiveness is limited and needed for informed policy decisions on adoption and scale up. We used a decision-tree analytic modelling approach, time-and-motion study, and routine data to estimate the potential impact of implementing stool-based TrueNat testing for the diagnosis of pulmonary tuberculosis in children within Nigerian primary healthcare settings on healthcare outcomes, resource use, health system costs, and cost-effectiveness relative to the standard of care (SoC). The cost per test was $13.06 (standard deviation; $0.77) for TrueNat and $16.25 (standard deviation; $1.34) for Xpert. For every 100 children with presumptive tuberculosis, the stool-based TrueNat testing intervention was projected to increase case detection rate by 2 (95% uncertainty interval [UI 0–6]) cases and bacteriological confirmation by 21% (95% UI 11–32). Diagnoses at primary health centres (PHC)

**Data availability statement:** All code and data to reproduce this analysis are publicly available on GitHub https://github.com/nmafirakureva/TruenatModel.

**Funding:** This work was made possible through the support of the United States Agency for International Development (USAID), Africa/SD and Global Health, under the terms of the Infectious Disease Detection and Surveillance contract GS00Q14OADU119. The USAID Country Team in Nigeria was involved in the design and implementation of the study and in producing the manuscript. The funders had no role in study design, data collection and analysis, decision to publish, or preparation of the manuscript. Views expressed are not necessarily those of USAID or the United States government. APC funding from University of Sheffield Institutional Open Access Fund.

**Competing interests:** The authors have declared that no competing interests exist.

would increase by 22% (95% UI 11–32), averting 1 (95% UI 0–2) deaths and 15 (95% UI -4–41) discounted DALYs. Although resource use and health system costs increased by $2,682 (95% UI 1,039-4,731) per 100 children, the incremental cost-effectiveness ratio of $183 per DALY averted suggests cost-effectiveness at thresholds of 0.5×GDP per capita. Implementing stool-based TrueNat testing has potential to increase access and reduce direct health system costs associated with the diagnosis of pulmonary tuberculosis in children in routine health care settings. Such an approach is likely to represent a good value for money compared to SoC.

## Introduction

Tuberculosis (TB) remains a significant cause of illness and death among children globally [1,2]. Previous modelling work showed that the majority of these deaths (96%) occur in children who do not receive treatment [2], largely due to underdiagnosis or underreporting [3]. Finding children with active tuberculosis disease and putting them on appropriate treatment is essential to avoid or reduce these deaths. However, the diagnosis of tuberculosis in children is limited by children's low bacillary load, difficulty in obtaining suitable diagnostic samples, and the presence of non-specific symptoms [4–6]. Additionally, existing diagnostic approaches are not child specific or are not available where they are needed the most [7,8].

For a long time, sputum, which is challenging to obtain especially from young children, has been the main specimen used for the diagnosis of pulmonary tuberculosis. Challenges in obtaining sputum in children necessitated using (semi-)invasive methods such as nasogastric aspiration and sputum induction. However, these methods are painful and can be stressful for children and caregivers and sometimes require hospitalisation. These methods are not available at all primary healthcare (PHC) facilities in tuberculosis endemic areas, where sick children usually first seek care. The WHO has recommended the use of Xpert MTB/RIF and XpertMTB/RIF Ultra (Cepheid, Sunnyvale, United States of America [USA]), hereinafter referred to as Xpert, on stool samples as an initial diagnostic test for detection of tuberculosis and resistance to rifampicin in children with signs and symptoms of pulmonary tuberculosis since 2020 [9,10]. The Simple One Step (SOS) stool processing method, developed by KNCV in collaboration with the Ethiopian Public Health Institute [11], was preferred for its simplicity and potential cost-effectiveness [12,13]. Previous modelling work showed that Xpert-Ultra testing on stool for the diagnosis of tuberculosis in children at near point of care is cost effective and could potentially increase access to bacteriological confirmation of tuberculosis and reduce mortality [14].

Nigeria bears one of the highest burdens of tuberculosis globally, ranking first in Africa and sixth worldwide, contributing approximately 4.6% of the global tuberculosis burden [1]. In 2023, children aged 0–14 years accounted for 10% of the 367,250 tuberculosis cases reported (from an estimated incidence of 499,000) in Nigeria [1,2]. However, TB notifications in children remain low, primarily due to under-diagnosis, especially among those under 5 years old. This is particularly concerning as tuberculosis tends to be more severe in children under 15, with the highest mortality observed in those under 5 years [1,2].

Nigeria introduced stool-based Xpert testing as an alternative to sputum samples for diagnosing tuberculosis in children in 2020 in selected states. This was initially limited to reference laboratories due to complex stool processing methods and was scaled-up country-wide in 2021. A recent study assessed the potential for decentralisation and the impact of stool-based Xpert testing using the SOS method in 14 states across Nigeria [15]. Findings from this study indicated that between October 2020 and September 2023, a total of 50,774 children were tested using stool samples, with *Mycobacterium tuberculosis* (MTB) detected in 2,440 (4.8%). However, another recent study showed limited availability of Xpert testing at primary healthcare facilities [16] in Nigeria, thus limiting this potential.

Since 2020, WHO has recommended using TrueNat MTB, MTB Plus and MTB-RIF Dx assays, hereinafter referred to as TrueNat, developed by Molbio Diagnostics (India), as initial diagnostic tests for tuberculosis in adults and children with signs and symptoms of pulmonary disease [17,18]. The TrueNat assays run on portable battery-operated devices, with minimal operational requirements and potential to be utilised as a point-of-care test in resource limited settings. Leveraging these features that make TrueNat ideal for placement within peripheral healthcare facilities and possibility of using stool samples should increase access to rapid diagnostic testing and bacteriological confirmation of tuberculosis in children. A pilot implementation of stool testing using TrueNat in routine settings was done in Nigeria during the first quarter of 2024 and the findings should become available soon. However, evidence on the health impact and cost-effectiveness of using stool on TrueNat platforms as point-of-care tests is limited [18], but urgently required to inform decisions on implementation and scale-up in routine healthcare systems. As a follow-on to this pilot implementation study, we modelled the potential impact and cost-effectiveness of implementing a stool-based TrueNat testing approach for the diagnosis of pulmonary tuberculosis in children in routine health care settings in Nigeria in comparison to standard of care.

## Methods

### Patient care pathways

**Standard of care.** Care pathways for the diagnosis of tuberculosis in children aged 0–14 years attending healthcare services under the standard of care were developed based on national guidelines [19] and discussion with country experts (see Fig 1). Children present at primary healthcare (PHC) or hospital level where they may be screened for tuberculosis symptoms and those presumed to have tuberculosis are further evaluated for active disease. At PHC, a proportion of these children may be referred to higher-level facilities immediately without tuberculosis investigations due to the presence of danger signs (defined as signs and symptoms that indicate severe illness or an urgent health problem) [20]. Evaluation for active tuberculosis disease may involve clinical assessment only with/without chest x-ray or clinical assessment with/without chest x-ray and bacteriological assessment depending on clinical need and test availability. Bacteriological assessment mainly involves Xpert on respiratory and stool samples or TrueNat on sputum only or TB LAMP (loop-mediated isothermal amplification) test on sputum specimens [19]. The lateral flow urine lipoarabinomannan (LF-LAM) assay is used for eligible children living with human immunodeficiency virus (HIV). In general, the capacity for bacteriological assessment for children is limited at PHC (and some hospitals) due to unavailability of tests or inability of children to provide appropriate samples required for testing.

Based on the assessment received, the outcomes can be diagnosis of tuberculosis (clinical or bacteriological) or no tuberculosis diagnosis. Tuberculosis diagnoses are further categorised into drug-susceptible (DS-TB) and drug-resistant (DR-TB) based on test results and assessment for risk factors for DR-TB. A proportion of children not initially receiving a tuberculosis diagnosis at PHC may be referred to a higher-level facility for further investigations and management. Children not meeting the criteria for tuberculosis diagnosis are asked to return to the facility for reassessment in 1–2 weeks if there are no improvements. Children diagnosed with active tuberculosis disease are initiated on anti-tuberculosis treatment using directly observed therapy at home or in the health facility. The selection of treatment regimen is based on disease severity, site, and drug susceptibility in accordance with the National TB guidelines [19].

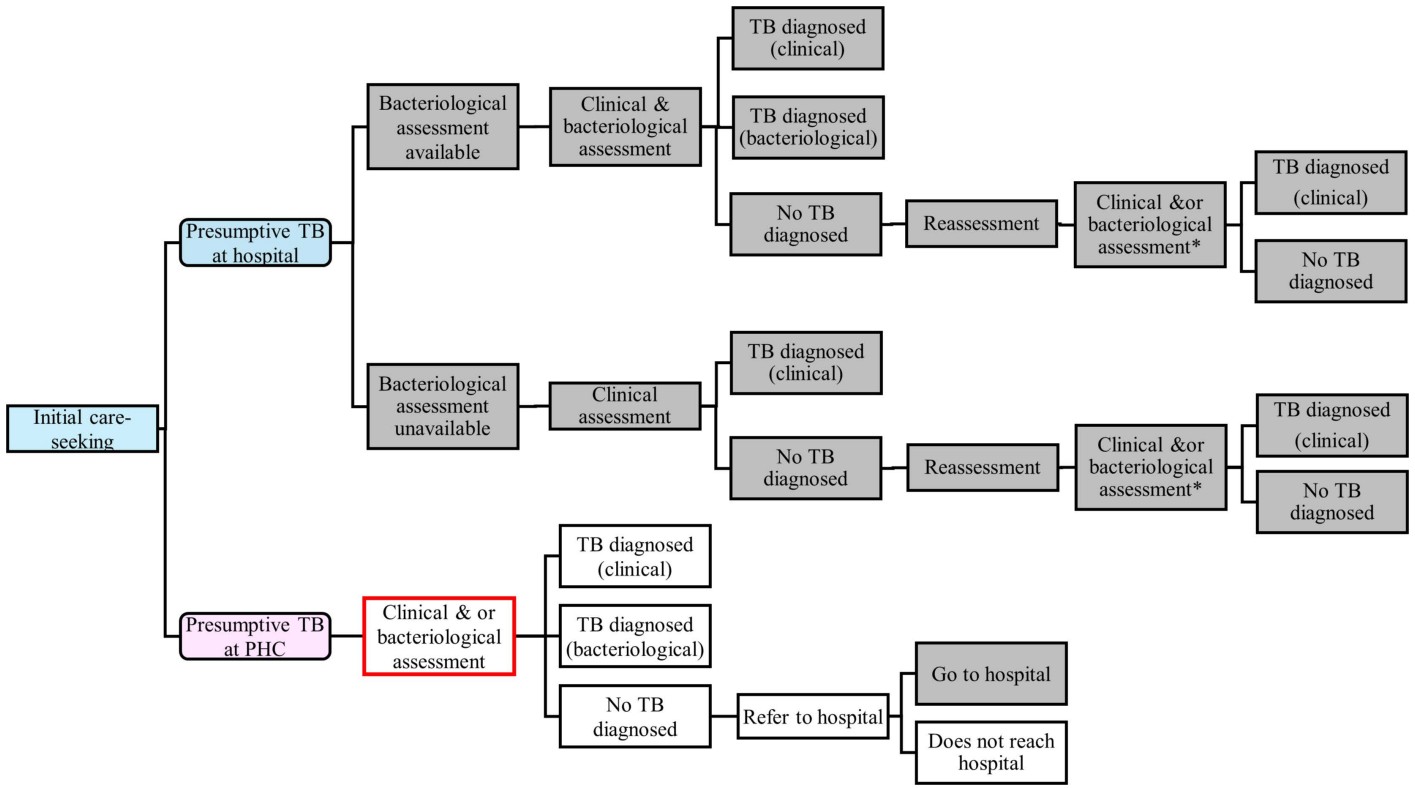

**Fig 1. Simplified diagram showing the pathways of care for the management of children with symptoms suggestive of tuberculosis (presumptive TB).** PHC = primary health centre, TB = tuberculosis. Boxes shaded in grey show activities undertaken at hospital level. Unshaded boxes show activities undertaken at PHC. The box with red outline shows where the stool-based TrueNat for bacteriological testing using the SOS method will be introduced under the intervention.

**Intervention.** The intervention was conceptualised as introducing stool-based TrueNat for bacteriological testing using the SOS method of children in PHC facilities where there is currently no access to bacteriological assessment using Xpert directly or through sample referral. Under conditions of limited or no access to Xpert testing, the intervention would increase the fraction of children with a bacteriological test result at PHC and reduce referrals to higher healthcare level. Reduction in referrals of children to higher level facilities would subsequently/possibly reduce referral lost-to-follow-up.

## Costing approach

**Cost data collection.** Data to inform resource use and costs associated with implementing a stool-based TrueNat testing approach were collected from a sample of laboratory facilities and literature. Data on quantities/costs of laboratory reagents and consumables, laboratory equipment and capital assets (i.e., medical and non-medical equipment), training, sample transportation, overhead (utilities, furniture, administrative, and operational costs) and building/infrastructure costs related to implementing the tests were collected from 5 out of 6 TrueNat and 7 out of 28 Xpert laboratories involved in the Nigeria TrueNat Stool Routine Implementation Pilot study (hereinafter, the evaluation study). Staff times used for performing TrueNat and Xpert test activities were collected using a time and motion study (see below). This was valued using monthly staff salaries obtained from the 12 participating laboratories

Costs of some reagents/consumables (if unavailable) and medications were obtained from the Stop TB partnership's Global Drug Facility catalogue [21]. Clinical assessment costs were based on WHO-CHOICE estimates for outpatient

visits [22]. Anti-tuberculosis treatment costs were derived from public sources using an approach previously described [23]. This approach uses country data available from the WHO global Tuberculosis database and estimate treatment costs as the sum of National TB program expenditures, costs for inpatient and outpatient care and the cost of paediatric specific anti-tuberculosis drugs.

**Time and motion study.** A time and motion study was performed to estimate the length of time that staff spent on each test's activities. Activity timing forms (time sheets) specifically designed for the evaluation study were used to record the length of time used to prepare samples, initiate tests, validate and record test results. These activity timings data were collected from 6 out of 6 TrueNat (30 timesheets) and 18 out of 28 Xpert (81 timesheets) laboratories. Approximately 5 timesheets were completed for each laboratory. Participants included a wide range of laboratory staff with different experience levels who provided data voluntarily and no personal data were collected. More details are provided in S1 Appendix.

**Cost analysis.** Unit costs for activities involved in clinical assessments (including reassessments), bacteriological tests and anti-tuberculosis treatments were estimated using an ingredient-based costing approach. This involves multiplying quantities of individual ingredients/resources used in an activity by their respective costs and then summing the resource costs to estimate the total cost for the activities. For example, the total cost for each test was calculated as the sum of costs of staff time, equipment use, cartridges, reagents, consumables (sample and test), sample transportation, building space and overheads. Similarly, the total treatment cost was calculated as the sum of costs of outpatient visits, inpatient stay, laboratory monitoring tests and medications. All costs were estimated and reported in 2024 United States dollar prices. A 3% annual discount rate was applied to annuitize equipment costs.

## Modelling approach

A decision-tree analytic modelling approach was used to evaluate the health impacts, healthcare system resource use and costs, and cost-effectiveness of the intervention in comparison to the standard of care approach for the diagnosis of tuberculosis in children. The decision-tree model was structured to represent the clinical care pathways shown in Fig 1. The model captured the cascade of care for children presenting to healthcare facilities, presumed to have tuberculosis and were subject to further evaluation for active tuberculosis disease. Cascade steps before a child is presumed (presentation and screening for symptoms) to have tuberculosis were included for completeness but were assumed to remain the same under the standard of care and intervention. The referral endpoints from PHC to higher level were modelled by adding an identical hospital care pathway to follow the path for referral from PHC level. This accounts for the flow of patients needing more specialised care or diagnostic services that cannot be provided at the PHC level. The intervention was modelled by replicating the entire pathway and increasing the proportion of children with access to bacteriological testing at PHC, enabled by the availability of TrueNat on stool samples. The model structure and all the associated parameters were discussed and agreed upon with in-country TB experts.

The probabilities of following different pathways through the decision tree were assumed to depend on these children's attributes: age (0–4 years or 5–14 years); HIV and antiretroviral treatment status (both positive or negative); and true tuberculosis status (bacteriologically-confirmable tuberculosis, bacteriologically-unconfirmable tuberculosis, no tuberculosis). Bacteriologically-confirmable tuberculosis refers to tuberculosis that would be bacteriologically positive under ideal circumstances and with all samples available. Bacteriologically-confirmable tuberculosis was further split into drug susceptible or drug-resistant tuberculosis.

Decision tree model probabilities were estimated based on data from past or ongoing studies in Nigeria, literature reviews and expert opinion/assumptions (see S1 Appendix for details). Age distribution of children was informed using data obtained from Institute of Human Virology, Nigeria (IHVN) tuberculosis programmes and the Nigeria TrueNat Stool Routine Implementation Pilot. The proportion of children initially seeking care from PHC, 82% (72–92%), was based on data from the 2018 Nigeria Demographic and Health Survey (2018 NDHS) [24]. For simplicity, all children presenting were assumed to be screened for tuberculosis symptoms, with a proportion of those screened being presumed to

have tuberculosis. This proportion was informed by the diagnostic accuracy of using symptoms (≥ 1 of cough, fever, or decreased playfulness) for screening of pulmonary tuberculosis in child contacts [25]. Country-specific data on the prevalence of HIV among children aged 0–14 years seeking care in healthcare facilities (2.4% (1.8-3.2%)) was used in the model [26]. UNAIDS data for Nigeria [27], was used to inform antiretroviral coverage (29% (25–32)).

Availability of bacteriological tests (particularly Xpert) at primary care and hospital level was informed by discussions with country experts and coverage data from Odume et al. [16]. Under the intervention, we assumed similar Xpert coverage as under the standard of care but an increased coverage of TrueNat at PHC matching that of Xpert at hospital level (20–100%) [16]. The proportion of children tested was modelled as depending on the possibility of obtaining a sample. The proportion of children able to provide a sputum sample was based on our previous analyses (~30% for 5–14 years and ~5% for 0–4 years) [14,28]. Diagnostic accuracy for Xpert on respiratory and stool samples for pulmonary tuberculosis in children was informed by data from recent systematic reviews [29]. Diagnostic accuracy data for TrueNat on stool samples for pulmonary tuberculosis in children is currently not available. We assumed the recently reported diagnostic performance of TrueNat on respiratory samples in comparison to microbiological reference standard as a proxy for the accuracy of TrueNat on stool samples: sensitivity of 57.1 (48, 65.9) and specificity of 92 (89.2, 94.2) [30]. The performance of Xpert on stool has been shown to be comparable to that on sputum [31]. The proportion of bacteriologically-confirmable tuberculosis that was drug- resistant was assumed to be 3% based on IHVN tuberculosis programme data. Data on referral loss to follow-up and 7/14-day loss to follow up was not available and was assumed to be similar to rates assumed in previous studies (around 20% (0–40%) [14,28].

Active tuberculosis disease outcomes were modelled using previously described approaches [2,14], by applying published meta-analytic mortality risk estimates specific to first-line treatment stratified by age, HIV and antiretroviral therapy status [32]. Life expectancy data for Nigeria obtained from United Nations population estimates [33] were used to calculate the average age-specific life-years lost due to premature mortality over a lifetime horizon (with and without 3% discounting) [34]. The estimated life-years lost were used to approximate disability-adjusted life-years (DALYs), neglecting the contribution of morbidity (consistent with previous modelling work which showed negligible impact) [35].

Estimated unit costs associated with resource use at each cascade of care steps were modelled as following gamma distributions with means and variances, and accumulated to produce total mean costs for each pathway. All costs were assumed to accrue in the present, with no discounting applied.

Whenever feasible, model parameters were considered uncertain and described using suitable probability distributions. A probabilistic sensitivity analysis-based approach was used to calculate all results by applying many samples (1,000) of all inputs for probabilistic uncertainty analysis, with means and 95% quantiles (uncertainty intervals [UIs]) reported.

## Health economic outcomes

For every 100 children seeking care with presumptive tuberculosis, we calculated the number of clinical assessments, bacteriological assessments, referrals, anti-tuberculosis treatments (ATTs), deaths, DALYs, and total costs, along with their changes under the intervention. Additionally, we calculated the percentage of true tuberculosis cases receiving ATT, ATT cases that are bacteriologically confirmed, ATT initiated at PHCs, and ATT cases that are false positives, as well as their changes under the intervention. ATT refers to the initiation of anti-TB drugs for individuals diagnosed with tuberculosis, whether confirmed or suspected; "true tuberculosis" refers to correctly diagnosed patients with tuberculosis, while "false tuberculosis" refers to those misdiagnosed with the disease despite not having it. The incremental cost-effectiveness ratios (ICERs) in terms of incremental cost per DALY averted and the probability of the intervention being cost-effective at different cost-effectiveness thresholds using a health-system perspective was also calculated and presented. With no defined cost-effectiveness threshold for Nigeria, we assumed an illustrative threshold of 0.5 × gross domestic product (GDP) per capita, aligning with recent reports suggesting thresholds lower than GDP per capita [36–38]. ICERs falling below 0.5 × GDP per capita suggest that the intervention is cost-effective against this threshold. However, the choice of threshold is ultimately a matter for policy makers. The whole analysis and reporting of results complies

with the Consolidated Health Economic Evaluation Reporting Standards (CHEERS 2022) reporting guidelines [39]. This analysis was not pre-specified in a formal health economic analysis plan but was included in the protocol for the *Routine evaluation of stool-based testing to diagnose tuberculosis using TrueNat.*

### Sensitivity analysis

The impact of the following modelling assumptions on cost-effectiveness outcomes were assessed in sensitivity and scenario analyses: (1) we applied alternative discount rates of 0% and 5% for the life-years and DALYs; (2) we considered universal availability of TrueNat as the bacteriological test at peripheral healthcare facilities under the intervention as opposed to limited coverage; (3) we applied a lower proportion (40%) of children initially presenting to primary healthcare facilities.

### Ethical considerations

The protocol for the *Routine evaluation of stool-based testing to diagnose tuberculosis using the TrueNat platform in comparison to Xpert Ultra on stool in Nigeria (DIGESTER STUDY),* which included this economic analysis, was reviewed and approved by the Osun State Health Research Ethical Committee (OSHREC) (REF: OSHREC/PRS/569T/504 dated 05 February 2024) and Oyo State Ministry of Health Research Ethics Committee (HREC) (REF: NHREC/ OYOSHRIEC/10/11/22 dated 02 February 2024).

### Patient and public involvement

Patients or the public were not involved in the design, or conduct, or reporting of this analysis.

## Results

The estimated unit costs applied in the model are shown in Table 1, with additional details provided in S1 Appendix. Of note, the unit cost for TrueNat ($13.10) was $3.00 less than that of Xpert ($16.20) test, despite larger staff-time in managing the additional steps involved in performing TrueNat tests. The difference in test costs was largely driven by the cost of equipment and reagents and consumables.

Projected changes in resources use, costs, and outcomes are shown in Table 2. The implementation of stool-based TrueNat testing for the diagnosis of pulmonary tuberculosis in children in primary health care settings in Nigeria was

**Table 1. Unit costs for tuberculosis evaluation and treatment.**

| Parameter description | PHC | Hospital |
|---|---|---|
| TB symptom screening | 3.75 (1.61) | 3.89 (1.64) |
| TB evaluation | 3.75 (1.61) | 3.89 (1.64) |
| Sample referral | 5.99 (0) | 5.99 (0) |
| GeneXpert | 16.25 (1.34) | 16.25 (1.34) |
| TrueNat | 13.06 (0.77) | 13.06 (0.77) |
| Treatment for drug sensitive TB (0–4 years) | 164.06 (31.82) | 164.06 (31.82) |
| Treatment for drug sensitive TB (5–14 years) | 186.83 (31.86) | 186.83 (31.86) |
| Treatment for drug resistant TB (0–4 years) | 6832.9 (248.17) | 6832.9 (248.17) |
| Treatment for drug resistant TB (5–14 years) | 12468 (265.75) | 12468 (265.75) |
| ATT initiation | 3.75 (1.61) | 3.89 (1.64) |
| ATT follow-up | 3.75 (1.61) | 3.89 (1.64) |

Costs are presented as mean (standard deviation) in 2024 United States dollars (US$). ATT = anti-tuberculosis treatment, PHC = primary health care, anti-tuberculosis treatment, TB = tuberculosis.

**Table 2. Healthcare resource use, health outcomes, costs & cost-effectiveness of the intervention in comparison to standard of care.**

| Quantity per 100 children with presumptive TB (unless stated) | Standard of care | Intervention | Increment |
|---|---|---|---|
| **Health-care resource use** | | | |
| Prevalence of true TB | 28 (16–40) | 28 (16–40) | 0 (0–0) |
| Total assessments | 192 (160–220) | 167 (145–188) | −25 (−44–−4) |
| Percent assessments at PHC† | 44 (33–54) | 68 (50–84) | 24 (11–38) |
| Bacteriological assessments | 50 (29–71) | 80 (63–93) | 30 (5–54) |
| Percent bacteriological assessments at PHC† | 8 (2–21) | 70 (43–89) | 61 (35–81) |
| Referrals to hospital | 48 (28–65) | 15 (4–32) | −32 (−50–−14) |
| Anti-tuberculosis treatments (ATT) | 38 (29–47) | 40 (32–49) | 2 (0–6) |
| Percent ATT initiated at PHC† | 48 (32–61) | 70 (49–87) | 22 (11–32) |
| Percent of true TB receiving ATT† | 82 (76–88) | 83 (77–87) | 0 (−3–5) |
| Percent of ATT bacteriologically confirmed† | 4 (2–8) | 25 (15–36) | 21 (11–32) |
| Percent of ATT false-positive† | 41 (26–59) | 44 (29–61) | 3 (1–6) |
| **Health outcomes** | | | |
| Tuberculosis diagnoses | 39 (30–49) | 42 (33–52) | 2 (0–6) |
| Percent of TB at PHC† | 48 (32–62) | 70 (50–87) | 22 (11–32) |
| Percent of TB bacteriologically confirmed† | 4 (2–8) | 25 (15–36) | 21 (11–32) |
| Deaths | 20 (17–24) | 19 (16–23) | −1 (−2–0) |
| Disability-adjusted life years (DALYs) | 1184 (979–1406) | 1150 (956–1366) | −34 (−95–10) |
| Discounted DALYs | 507 (419–602) | 492 (409–585) | −15 (−41–4) |
| **Health systems costs** | | | |
| Cost (2024 US$) | 8731 (6720–11191) | 11414 (8910–14728) | 2682 (1039–4731) |
| **Cost-effectiveness analysis** | | | |
| ICER (Cost per DALY averted) | | | 183 |

†Indicates percentages calculated using different denominators. Data are presented as means and 95% uncertainty interval from probabilistic sensitivity analysis unless otherwise stated. Uncertainty in cost-effectiveness is presented in Fig 2. All costs are presented in 2024 United States dollars (US$). The incremental cost-effectiveness ratio (ICER) is presented as US$ per discounted disability-adjusted life year (DALY) averted. PHC = primary healthcare centre, TB = tuberculosis.

projected to result in an additional 2 (95% UI 0–6) tuberculosis cases detected under the intervention. The intervention was projected to result in a 21 (95% UI 11–32) percentage point increase in the proportion of children with bacteriologically confirmed tuberculosis. The proportion of children diagnosed with tuberculosis at PHC increased from 48% (95% UI 32–62) under SoC to 70% (95% UI 50–87) under the intervention. The intervention effect on tuberculosis case detection rate resulted in an estimated 1 (95% UI 0–2) deaths avoided and 15 (95% UI -4–41) discounted DALYs averted.

These improvements were achieved by increasing the level of resources used under the intervention. Although for every 100 children seeking care with presumptive tuberculosis, the overall number of assessments decreased by 25 (95% UI 4–44) from 192 (163–220) under SoC to 167 (145–188) under the intervention, bacteriological assessments increased by 30 (95% UI 5–54) assessments from 50 (95% UI 29–71) to 80 (95% UI 63–93). The intervention would increase the proportion of assessments at PHC by 24 (95% UI 11–38) percentage points from 44% (95% UI 33–54) to 68% (95% UI 50–84) and the proportion of bacteriological assessments at PHC by 61 (95% UI 35–81) percentage points from 8% (95% UI 2–21) to 70% (95% UI 43–89). The model projected an increase in the number of children starting anti-tuberculosis treatments (ATT) from 38 (95% UI 29–47) under the standard of care to 40 (95% UI 32–49) under the intervention; an increase of 2 (95% UI 0–6).

The projected increase in the number of assessments and anti-tuberculosis treatments would increase health system costs by $2,682 (95% UI 1,039 - 4,731), from $8,731 (95% UI 6,720–11,191) under SoC to $11,414 (8,910–14,728) under the intervention.

The incremental cost-effectiveness ratio (ICER) for implementing the intervention in comparison to SoC was estimated $183 per DALY averted. The intervention would be considered cost-effective assuming a cost-effectiveness threshold of 0.5×GDP per capita (equivalent to $810 in Nigeria) per DALY averted. Age disaggregated results are presented in the S1 Appendix. Fig 2 shows the probability of the intervention being cost-effective compared to SoC over a range of willingness-to-pay thresholds (representing decision uncertainty). The probability of the intervention being cost-effective exceeded 50% at a cost-effectiveness threshold of $203 per DALY averted. The maximum probability of the intervention being cost-effective (~100%) would be achieved at a cost-effectiveness threshold of $493 per DALY averted. The distribution of the incremental costs and disability-adjusted life-years averted by the intervention are shown on Fig 3.

In the sensitivity analyses, assumptions on the discount rate applied on life years had the largest impact on model results, although the intervention remained cost-effective (see S1 Appendix). The ICERs dropped to $67 per DALY averted with no discounting and increased to $236 per DALY averted with a 5% discount rate.

## Discussion

This analysis showed that adding stool-based TrueNat testing for the diagnosis of pulmonary tuberculosis in children in routine health care settings in Nigeria has potential to increase bacteriological assessments and confirmation at primary healthcare level. Although these improvements were achieved by increasing the overall level of resources used under the

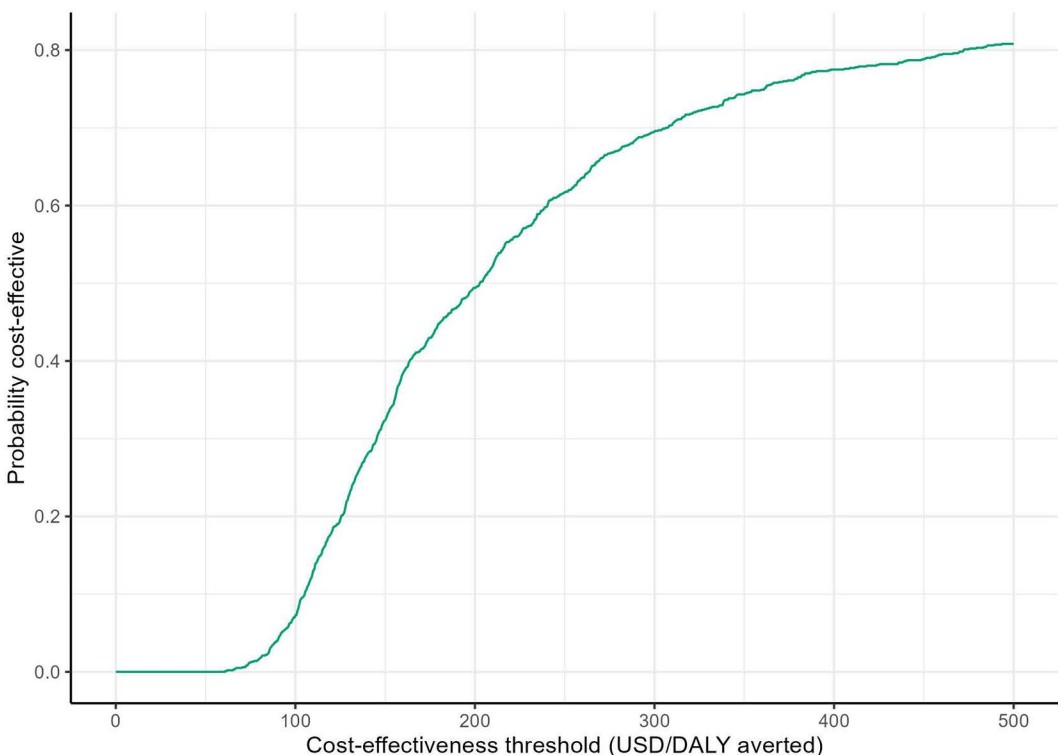

**Fig 2. Cost-effectiveness acceptability curves for the intervention in comparison to the standard of care.** The figure shows the probability that an intervention is cost-effective (y-axis) based on the proportion of simulations in which the comparison of the intervention to the standard of care falls below the cost-effectiveness threshold (y-axis).

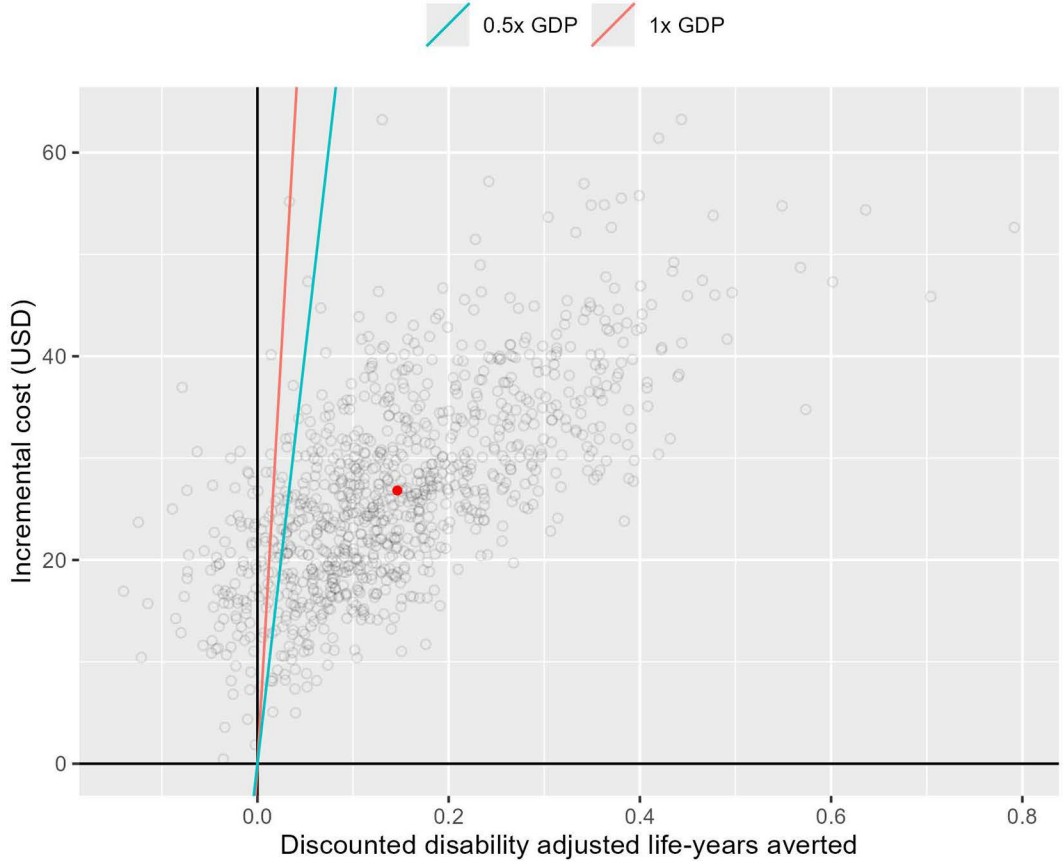

**Fig 3. Cost-effectiveness plane showing the differences in costs (y-axis) and disability-adjusted life-years (DALYs, x-axis) of using the stool-based TrueNat for bacteriological testing using the simple one-step method for diagnosis of tuberculosis in children, compared with standard of care from 1000 simulations.** Cost-effectiveness threshold (CET) lines show the cost-effectiveness (willingness to pay) threshold based on 1X GDP per capita (red line) or 0.5X GDP per capita (green line) in each country. The red dot represents the mean incremental costs and DALYs averted. ICER, incremental cost effectiveness ratio; CET, cost-effectiveness threshold; GDP, gross domestic product per capita, USD = United States dollar.

intervention, the 1 (95% UI 0–2) deaths and 15 (95% UI -4–41) DALYs averted per 100 children seeking care with presumptive tuberculosis would result in the intervention being cost-effective (ICER; $183 per DALY averted) relative to SoC, assuming cost-effectiveness thresholds of about 0.5X the GDP per capita for Nigeria.

Although the overall number of assessments decreased by 25 (4–44) under the intervention, bacteriological assessments increased by 30 (5–54) assessments from 50 (29–71) to 80 (63–93). In particular, the intervention would increase the proportion of assessments at PHC from 44% (33–54) to 68% (50–84) and the proportion of bacteriological assessments by 61 (35–81) percentage points from 8% (2–21) to 70% (43–89). This is important in programmatic settings as most current data suggest that the largest proportion of children initially seek care at PHC [14,24,28,40]. These lower level facilities have perennially been known to have limited services [40] or capacity for diagnosing tuberculosis [15,16]. The overall number of assessments decreased due to reduced upward referral of children initially assessed at PHC and requiring further assessment at a higher level. Avoiding referrals is potentially important for reducing delays in the care pathway of sick children and ensuring timely diagnosis or exclusion of tuberculosis to enable early provision of appropriate care. This is likely to improve outcomes for these children who are particularly more vulnerable to the effects of active tuberculosis disease. Additionally, this has potential for reducing the health system as well as patient costs.

Nigeria introduced the TrueNat MTB Plus and RIF assays in 2021 and has already incorporated them into national tuberculosis guidelines [41], paving way for national roll-out. Effectiveness and cost-effectiveness evidence will help inform decisions on wider implementation. This study serves as an early example of implementing a stool-based TrueNat tuberculosis diagnostic approach, providing valuable insights for other countries seeking to adopt similar methods. Given the current global need for innovative and accessible diagnostic solutions for tuberculosis in children, insights from this study are timely and valuable.

Our analysis leveraged the Nigeria TrueNat Stool Routine Implementation Pilot (first quarter 2024) to gather primary data to estimate the unit costs per test for TrueNat and Xpert in Nigeria. This was based on a time and motion study and data collection from a representative sample of laboratories implementing the two tests using stool samples. Therefore, our analysis did not rely on literature values which may vary by context depending on different operational conditions. Although Xpert test costs are widely available, the costs of TrueNat testing are currently limited to a handful of studies [42,43], which show a substantial variation in the costs highlighting the need for context-specific estimates.

The analysis is not without some limitations. This was a modelling study using several assumptions based on best available evidence in the absence of empirical evidence. For example, diagnostic accuracy data for TrueNat specifically on stool samples for pulmonary tuberculosis in children is not yet available. In the absence of these data, we relied on recently reported diagnostic performance data on respiratory samples in comparison to microbiological reference standard [30] as a proxy for the accuracy on stool samples. Diagnostic accuracy of Xpert on stool and respiratory samples has been shown to be comparable in a systematic review and meta-analysis focusing on children living in African countries [31]. Assuming a lower accuracy for TrueNat on stool samples than what is reported for sputum samples could potentially reduce the cost-effectiveness of the intervention. Therefore, the analysis may require updating as new accuracy data becomes available.

The current analysis does not provide evidence on affordability of the new diagnostic approach, which is also an important consideration in addition to cost-effectiveness. A detailed budget impact assessment was not in scope of the current analysis but would be required to estimate the additional costs that the public health system would incur to achieve the health improvements we projected in our model. To put it into context, the approximately 40 TrueNat Duo systems currently being implemented in Nigeria would require an upfront investment of $US 560,000 in equipment costs only (assuming $14,000 per machine). Additional costs for test kits, training and logistics would further increase the required budget. A careful budget impact analysis would be required to determine the costs for wider national roll-out across the 36 states and over 30,000 PHC facilities [44].

Stool-based TrueNat testing should be considered as an alternative approach for the diagnosis of pulmonary tuberculosis in children, particularly in routine healthcare settings where other bacteriological tests are not readily available. However, it is important to recognise that bacteriological testing, including TrueNat, can have a low negative predictive value, especially in young children. As such, the introduction of this technology should be accompanied by comprehensive training for healthcare professionals, ensuring that clinical assessment remains central to decision-making, particularly when negative test results are obtained.

Moreover, policymakers must carefully evaluate the full spectrum of costs associated with the implementation of TrueNat testing. This includes not only the procurement of equipment, reagents, and the establishment of supply chain logistics, but also the cost of training healthcare providers, developing or updating clinical guidelines, and improving infrastructure to support the integration of this technology. Additionally, consideration should be given to ongoing maintenance costs, the potential need for additional resources, and the long-term sustainability of such interventions within the healthcare system.

Despite some limitations, this analysis offers promising insights for broader application. In settings where sputum collection in children is challenging, stool-based testing provides a non-invasive alternative. The stool-based TrueNat approach has potential for scaling in other countries with similar barriers, enhancing access to tuberculosis diagnostics for

children and contributing to global TB elimination efforts. Future research across diverse regions will further validate these findings and help tailor the approach to different epidemiological and healthcare contexts.

The implementation of stool-based TrueNat testing at primary healthcare level has potential to increase access and reduce direct costs associated with the diagnosis of pulmonary tuberculosis in children in routine health care settings. The intervention is likely to represent a good value for money compared to standard of care. However, the impact of wider implementation of such an intervention on the public healthcare system budgets needs careful assessment to determine its affordability.

## Supporting information

**S1 Appendix. Supplementary methods and results.**
(DOCX)

## Acknowledgments

We sincerely appreciate the dedication of the healthcare workers and laboratory staff across all study sites for their efforts in enrolling children into this study. Our deepest gratitude goes to the children and their caregivers for their invaluable contributions to advancing scientific knowledge.

We also extend our thanks to the following individuals and organizations for their expert guidance during protocol development: Wayne van Gemert and Lucy Mupfumi (Stop TB Partnership); colleagues from the National Tuberculosis, Leprosy and Buruli Ulcer Control Programme; the Osun and Oyo State Tuberculosis, Leprosy and Buruli Ulcer Control Programmes; United States Agency for International Development (USAID); Institute of Human Virology, Nigeria (IHVN); and Infectious Disease Detection and Surveillance (IDDS) for their valuable input to the study. For the purpose of open access, the author has applied a Creative Commons Attribution (CC BY) licence to any Author Accepted Manuscript version arising.

## Author contributions

**Conceptualization:** Nyashadzaishe Mafirakureva, Eveline Klinkenberg, Peter J. Dodd.

**Data curation:** Nyashadzaishe Mafirakureva, Olabamiji Jamiu Olayinka.

**Formal analysis:** Nyashadzaishe Mafirakureva, Peter J. Dodd.

**Methodology:** Nyashadzaishe Mafirakureva, Peter J. Dodd.

**Resources:** Olabamiji Jamiu Olayinka.

**Supervision:** Peter J. Dodd.

**Visualization:** Nyashadzaishe Mafirakureva.

**Writing – original draft:** Nyashadzaishe Mafirakureva, Eveline Klinkenberg, Peter J. Dodd.

**Writing – review & editing:** Nyashadzaishe Mafirakureva, Olugbenga Kayode Daniel, Olabamiji Jamiu Olayinka, Kingsley Chinedum Ochei, Eveline Klinkenberg, Austin Ihesie, Debby Nongo, Rupert Amanze Eneogu, Andwele Mwansasu, Emeka Uga Elom, Agbaje Vivian Aderonke, Patrick Sunday Dakum, Charles Olalekan Mensah, Oluwafemi Christopher Odola, Abiola Oladotun Olayemi, Emily Yemisi Faleye, Adekunle Omotoso Makinde, Peter J. Dodd.

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
