## [Decision Letter · Decision Letter 0]

2 Jan 2025

PGPH-D-24-02630

Evaluating the health impact, health-system costs and cost-effectiveness of using TrueNat on stool samples compared to usual care for the diagnosis of paediatric tuberculosis in primary care settings: a modelling analysis.

Dear Dr. Mafirakureva,

Thank you for submitting your manuscript to PLOS Global Public Health. After careful consideration, we feel that it has merit but does not fully meet PLOS Global Public Health’s publication criteria as it currently stands. Therefore, we invite you to submit a revised version of the manuscript that addresses the points raised during the review process.

We look forward to receiving your revised manuscript.

Kind regards,

Raquel Muñiz-Salazar, Ph.D.

Academic Editor

Journal Requirements:

Additional Editor Comments (if provided):

Both reviewers found the manuscript to be well-written, relevant, and methodologically robust, addressing an important and timely research question about the use of TruNat on stool samples for pediatric TB diagnosis. However, they provided constructive suggestions for improvement:

Key Suggestions for Improvement:

-Expand the introduction to include specific data on the burden of pediatric TB in Nigeria.

-Provide more details on the decision tree model, including a visual figure, and ensure variables reflect the Nigerian context.

-Clarify key terms such as True TB, false TB, and ATTs.

-Address inconsistencies in terminology, such as "proportion" vs. "number," and provide more information on the uncertainty of model estimates.

-Explain the observed discrepancy between the 29% increase in overall TB case detection and the 6% increase in bacteriologically confirmed TB cases.

-Strengthen the discussion by including more specific recommendations for policymakers and clinicians.

-Justify the choice of the cost-effectiveness threshold (0.5x GDP per capita).

-Provide details on the selection criteria for PHC facilities where the intervention was implemented, especially in the absence of bacteriological assessment using Xpert.

Reviewers' comments:

Reviewer's Responses to Questions

**Comments to the Author**

1. Does this manuscript meet PLOS Global Public Health’s publication criteria ? Is the manuscript technically sound, and do the data support the conclusions? The manuscript must describe methodologically and ethically rigorous research with conclusions that are appropriately drawn based on the data presented.

Reviewer #1: No

Reviewer #2: Yes

2. Has the statistical analysis been performed appropriately and rigorously?

Reviewer #1: N/A

Reviewer #2: Yes

3. Have the authors made all data underlying the findings in their manuscript fully available (please refer to the Data Availability Statement at the start of the manuscript PDF file)?

Reviewer #1: Yes

Reviewer #2: Yes

4. Is the manuscript presented in an intelligible fashion and written in standard English?

Reviewer #1: Yes

Reviewer #2: Yes

5. Review Comments to the Author

Reviewer #1: The manuscript is interesting, well-structured, and easy to follow. Here are some suggestions:

Clinical assessments are vital to TB diagnosis, and accurate clinical classification is necessary for subsequent evaluations. However, the model does not consider the proportion of cases that have the proper clinical classification.

Some of the variables will not be assumed or taken from other country. Example: The diagnostic accuracy of TruNat using stool might differ from that of TruNat using sputum; the loss to follow-up depends on the strength of the country's TB program, and data from other studies might not accurately represent Nigerian situations. The outcome of active TB considers age and HIV; nevertheless, this has to be taken from Nigeria.

The outcome variables are not clear, and please consider including definitions of terms. Examples: True TB, false TB, ATTs, etc.

While the result in Table 2 indicates mean, the terms proportion and number are used interchangeably throughout the main text.

The result showed the intervention increased TB case detection by 29%, but the bacteriologically confirmed TB increased only by 6%. How does the implementation of stool-based TureNat testing increase TB case detection more, but it brings minimal change to the bacteriologically confirmed TB?

Reviewer #2: Overall, this is a well-written and informative manuscript. The research question is important and timely, and the study is well-designed. The authors have used a robust modelling approach to evaluate the potential impact of using TrueNat on stool samples for pediatric tuberculosis diagnosis, and their findings are likely to be of interest to policymakers and clinicians in Nigeria and other countries with a high burden of TB.

Here are some specific comments:

• Strengths:

o The study addresses an important and timely topic.

o The study design is appropriate for the research question.

o The authors have used a robust modelling approach.

o The study findings are likely to be of interest to policymakers and clinicians.

Areas for improvement:

• Introduction:

o The introduction provides a good overview of the topic. However, it could be strengthened by providing more specific information on the burden of pediatric TB in Nigeria.

• Methods:

o The methods section is generally well-written. However, it could be improved by providing more detail on the decision tree model. For example, the authors could provide a figure of the decision tree.

• Results:

o The results section is clear and concise. However, it could be improved by providing more information on the uncertainty around the model estimates.

• Discussion:

o The discussion section is thoughtful and well-written. However, it could be improved by providing more specific recommendations for policymakers and clinicians.

Specific comments:

• Page 3, line 11: You state that the intervention was cost-effective at a threshold of 0.5x GDP per capita. Please provide more information on how this threshold was chosen.

• Page 4, line 21: You state that the intervention was implemented in PHC facilities where there was no access to bacteriological assessment using Xpert. Please provide more information on the selection criteria for these facilities.

6. PLOS authors have the option to publish the peer review history of their article (what does this mean? ). If published, this will include your full peer review and any attached files.

**Do you want your identity to be public for this peer review?** For information about this choice, including consent withdrawal, please see our Privacy Policy .

Reviewer #1: No

Reviewer #2: No

---

## [Decision Letter · Decision Letter 1]

29 Apr 2025

Evaluating the health impact, health-system costs and cost-effectiveness of using TrueNat on stool samples compared to usual care for the diagnosis of paediatric tuberculosis in primary care settings: a modelling analysis.

PGPH-D-24-02630R1

Dear Doctor Mafirakureva,

We are pleased to inform you that your manuscript 'Evaluating the health impact, health-system costs and cost-effectiveness of using TrueNat on stool samples compared to usual care for the diagnosis of paediatric tuberculosis in primary care settings: a modelling analysis.' has been provisionally accepted for publication in PLOS Global Public Health.

Best regards,

Raquel Muñiz-Salazar, Ph.D.

Academic Editor

Following a thorough review of the revised manuscript, we confirm that all reviewers' comments have been satisfactorily addressed.

We are pleased to inform you that the final decision is to accept your manuscript for publication.

Reviewer Comments (if any, and for reference):

Reviewer's Responses to Questions

**Comments to the Author**

1. If the authors have adequately addressed your comments raised in a previous round of review and you feel that this manuscript is now acceptable for publication, you may indicate that here to bypass the “Comments to the Author” section, enter your conflict of interest statement in the “Confidential to Editor” section, and submit your "Accept" recommendation.

Reviewer #2: All comments have been addressed

Reviewer #3: All comments have been addressed

2. Does this manuscript meet PLOS Global Public Health’s publication criteria ? Is the manuscript technically sound, and do the data support the conclusions? The manuscript must describe methodologically and ethically rigorous research with conclusions that are appropriately drawn based on the data presented.

Reviewer #2: Yes

Reviewer #3: Yes

3. Has the statistical analysis been performed appropriately and rigorously?

Reviewer #2: Yes

Reviewer #3: Yes

4. Have the authors made all data underlying the findings in their manuscript fully available (please refer to the Data Availability Statement at the start of the manuscript PDF file)?

Reviewer #2: Yes

Reviewer #3: Yes

5. Is the manuscript presented in an intelligible fashion and written in standard English?

Reviewer #2: Yes

Reviewer #3: Yes

6. Review Comments to the Author

Reviewer #2: (No Response)

Reviewer #3: This is a well written paper, technical but also likely accessible to those who may not be health economists.

7. PLOS authors have the option to publish the peer review history of their article (what does this mean? ). If published, this will include your full peer review and any attached files.

**Do you want your identity to be public for this peer review?** For information about this choice, including consent withdrawal, please see our Privacy Policy .

Reviewer #2: No

Reviewer #3: No
